# Application of Machine Learning to Predict COVID-19 Spread via an Optimized BPSO Model

**DOI:** 10.3390/biomimetics8060457

**Published:** 2023-09-28

**Authors:** Eman H. Alkhammash, Sara Ahmad Assiri, Dalal M. Nemenqani, Raad M. M. Althaqafi, Myriam Hadjouni, Faisal Saeed, Ahmed M. Elshewey

**Affiliations:** 1Department of Computer Science, College of Computers and Information Technology, Taif University, P.O. Box 11099, Taif 21944, Saudi Arabia; eman.kms@tu.edu.sa; 2Otolaryngology-Head and Neck Surgert Department, King Faisal Hospital, P.O. Box 11099, Taif 21944, Saudi Arabia; saraassiriiii@gmail.com; 3College of Medicine, Taif University, P.O. Box 11099, Taif 21944, Saudi Arabia; d.nemenqani@tu.edu.sa (D.M.N.); raad@tu.ed.sa (R.M.M.A.); 4Department of Computer Sciences, College of Computer and Information Science, Princess Nourah bint Abdulrahman University, P.O. Box 84428, Riyadh 11671, Saudi Arabia; 5DAAI Research Group, Department of Computing and Data Science, School of Computing and Digital Technology, Birmingham City University, Birmingham B4 7XG, UK; faisal.saeed@bcu.ac.uk; 6Faculty of Computers and Information, Computer Science Department, Suez University, Suez 43533, Egypt; ahmed.elshewey@fci.suezuni.edu.eg

**Keywords:** k-nearest neighbor, binary particle swarm optimization, random oversampling, random forest model, gradient boosting model, naive Bayes model

## Abstract

During the pandemic of the coronavirus disease (COVID-19), statistics showed that the number of affected cases differed from one country to another and also from one city to another. Therefore, in this paper, we provide an enhanced model for predicting COVID-19 samples in different regions of Saudi Arabia (high-altitude and sea-level areas). The model is developed using several stages and was successfully trained and tested using two datasets that were collected from Taif city (high-altitude area) and Jeddah city (sea-level area) in Saudi Arabia. Binary particle swarm optimization (BPSO) is used in this study for making feature selections using three different machine learning models, i.e., the random forest model, gradient boosting model, and naive Bayes model. A number of predicting evaluation metrics including accuracy, training score, testing score, F-measure, recall, precision, and receiver operating characteristic (ROC) curve were calculated to verify the performance of the three machine learning models on these datasets. The experimental results demonstrated that the gradient boosting model gives better results than the random forest and naive Bayes models with an accuracy of 94.6% using the Taif city dataset. For the dataset of Jeddah city, the results demonstrated that the random forest model outperforms the gradient boosting and naive Bayes models with an accuracy of 95.5%. The dataset of Jeddah city achieved better results than the dataset of Taif city in Saudi Arabia using the enhanced model for the term of accuracy.

## 1. Introduction

Abdelsalam, M. et al. [1] state that the COVID-19 pandemic was widely spread all over the world and that many countries all over the world were affected by the COVID-19 pandemic. COVID-19 has infected more than 147 million people around the world. The spread of the COVID-19 pandemic can through a symptomatic form or an asymptomatic form. Snuggs, S. et al. [2] demonstrated that due to the COVID-19 pandemic, many types of food were limited for patients who suffered from COVID-19. Galanakis, C. M. [3] illustrated that food systems are very efficacious with respect to human health since these systems effect psychological health. The CDC [4] demonstrated that COVID-19 infirmity was due to severe acute respiratory syndrome coronavirus 2 (SARS-CoV-2) that arose in Wuhan city in China at the end of 2019. This led to the beginning of the COVID-19 pandemic. Arias-Reyes, C. et al. [5] illustrated that for people residing at altitudes of more than 3000 m, the research clearly showed a decrease in the prevalence and impact of SARS-CoV-2 infection. The explanation for the global COVID-19 outbreak’s decreased intensity at a high altitude could be due to both environmental and physiological reasons. Castagnetto-Mizuaray, J. M. et al. [6] demonstrated that people believed that altitude could protect them against COVID-19 infection and death was not real; however, their experimental results illustrated that there was no influence of altitude on COVID-19 infection. Intimayta-Escalante, C. et al. [7] illustrated that there were some drawbacks to the study, and that the results were only applicable to the region in which it was conducted and could not be applied to people that live in other regions. Furthermore, The statistical analysis was carried out using data from COVID-19 positive cases; however, a large number of COVID-19 cases have not yet been verified through laboratory testing; In many nations, including Peru, it is impossible to accurately count all COVID-19 fatalities using national methods. Lin, E. M. et al. [8] illustrated that environmental circumstances may have a comparable impact on the prevalence of COVID-19 disease in Brazil. Deaths from COVID-19 have a strong correlation with low altitude and dense population. It has been revealed that high altitude could be a protective factor in its own right. Because there was little change in daylight over the study period, sunshine did not appear to have a substantial impact on COVID-19 outcomes. Segovia-Juarez, J. et al. [9] used data from the COVID-19 pandemic in Peru, where the first case was recorded in March 2020. There were 224,132 SARS-CoV-2 positive cases and 6498 deaths in June 2020. The incidence of COVID-19 infection was reduced at a high altitude, based on data from 185 provincial capitals with altitudes from 3 to 4342 m. Woolcott, O. O. et al. [10] illustrated that in Mexico and the United States, people who live above 2000 m have a greater total cumulative number of COVID-19 cases and a higher COVID-19-related mortality rate than those who live below 1500 m. In Mexico, altitude has been linked to COVID-19 deaths for patients younger than 65 years old. Shams, M. Y. et al. [11] proposed a model called the healthy artificial nutrition analysis (HANA) during the COVID-19 pandemic to maintain food for patients. Ferri, M. [12] presented that the prediction of COVID-19 samples was very vital by using deep learning of artificial intelligence (AI) and machine learning. Machine learning is a subset of AI that allows systems and algorithms to automatically learn without human intervention and gives computers the ability to make decisions regarding many problems in several fields such as finance, banking, and medicine, where the algorithms and systems improve based on the experience gained and without any traditional programming. In this work, an enhanced model for predicting COVID-19 samples in different regions of Saudi Arabia (high-altitude and sea-level areas) is proposed. The model uses several machine learning models, i.e., the random forest model, gradient boosting model, and naive Bayes model, as we will discuss in the next sections.

The remainder of this paper is organized as follows: Section 2 outlines the related work, Section 3 presents the materials and methods, Section 4 illustrates the experiments results and discussion, and Section 5 presents the conclusion and future work.

## 2. Related Work

De Castro, Y. et al. [13] used methods like polynomial regression (PR) combined with the polynomial kernel of degree 6 for COVID-19 prediction. Hao, Y. et al. [14] have used methods based on the Bertalanffy-Pütter (BP) model, logistic model, and Gompertz model to describe the growth of COVID-19. Their comparison results for the prediction of the cumulative number of confirmed diagnoses showed that the three models can better forecast COVID-19 evolution trends in the later stages of the epidemic. Yang, Z. et al. [15] used a logistic model. They also employed exponential models in their predictions. Adnan, M. et al. [16] predicted the overall spread of COVID-19 using the Artificial Neural Network (ANN) algorithm, as well as polynomial regression, SVM, and Bayesian Ridge Regression (BRR). Batista, A. et al. [17] trained and tested neural networks (NN), gradient boosting trees (GBDT), random forests (RF), logistic regression (LR), and support vector machines (SVM) for comparison. The best prediction was provided by SVM, with an AUC of 0.85. To predict the symptoms of infected patients, Sun, et al. [18] proposed the use of a Support Vector Machine (SVM)-based model and used the area under the receiver operating characteristic curve (AUROC) as a performance metric. They achieved 99.6% accuracy in the training phase and 97.57% accuracy in the testing phase. Salama, A. et al. [19] used artificial neural network (ANN), support vector machine (SVM), and linear regression (LR) models to predict the recovery estimation of COVID-19 patients. They concluded that SVM provided the best prediction accuracy, reaching 85%. Laatifi, M. et al. [20] illustrated a feature engineering machine learning (ML) method used to predict COVID-19 illness severity. This method is based on the Uniform Manifold Approximation and Projection (UMAP), a top-level data reduction method. They mainly used the following classifiers: AdaBoost classifier, random forest, XGB classifier, and extra trees, achieving 100% accuracy. However, they faced a challenge due to the limited data size. For the purpose of clustering low and high severity risk patients, Banoei, MM. et al. [21] used the statistical method SIMPLES for prediction and a Latent class analysis (LCA) for clustering. Their model had an AUC higher than 0.85. Iwendi, C. et al. [22] proposed a model to forecast the gravity of different cases, including recovery, outcome, and death, using a fine-tuned random forest boosted by the AdaBoost model. CSSE J. [23] utilized a dataset called the "novel Coronavirus 2019 dataset" and incorporated several features to achieve high accuracy, reaching 94% accuracy and an F1 Score of 0.86. In the field of COVID-19 mortality risk prediction, Pourhomayoun. et al. [24] utilized various machine learning (ML) algorithms, including support vector machine (SVM), decision tree (DT), logistic regression (LR), random forest (RF), and k-nearest neighbor (KNN). Their findings revealed that the ANN algorithm achieved the highest level of accuracy., at 89.98%. In another comparative study, Yadaw. et al. [25] employed an extreme gradient boosting (XGBoost) model, which outperformed LR, RF, and SVM. They used three features: age, minimum oxygen saturation, and healthcare setting. Moulaei, K. et al. [26] developed J48 DT, RF, KNN, multi-layer perceptron (MLP), naive bayes, XGBoost, and LR models using a laboratory-proven COVID-19 hospitalized patient dataset. Their results indicated that RF achieved the best accuracy at 95.03%. Hu, C. et al. [27] compared bagged flexible discriminant analysis (FDA), partial least squares (PLS) regression, LR, RF, and elastic net (EN) model. The AUC results demonstrated that LR, RF, and bagged FDA models performed well. Their final model used LR (with age, CRP level, lymphocyte count, and D-dimer level) and reached an 88.1% AUC. As an evolution of machine learning, deep learning (DL) has been widely used in the COVID-19 literature. Chae, S. et al. [28] proposed a DL framework based on the long short-term memory method, the autoregressive integrated moving average method (ARIMA), and ordinary least squares (OLS). Ezzat, D. et al. [29] proposed an approach called GSA-DenseNet121-COVID-19 using a hybrid Convolutional Neural Network (CNN), DenseNet121, with an optimization algorithm, the gravitational search algorithm (GSA). Their results indicated an accuracy of 98%. Deep neural networks (DNN), random forests (RF), and XGBoost were used by Yang. et al. [30] for diagnosis prediction. They used features including age, routine blood test results, and gender of patients were gathered from patients hospitalized to the Department of Infectious Diseases at the University Medical Center Ljubljana (UMCL), Slovenia. XGBoost performed the best, with a sensitivity of 81.9% and specificity of 97.9%.

## 3. Materials and Methods

In this paper, an enhanced model is used to predict COVID-19 samples for two datasets in Saudi Arabia. The proposed enhanced model is based on eight major steps, as demonstrated in Figure 1. There are eight steps used to develop the enhanced model, which are: (1) dataset collection, (2) utilizing the random oversampling method for data balancing, (3) applying the KNN imputation algorithm to impute missing data, (4) conducting data preprocessing, (5) employing the BPSO optimization algorithm for feature selection, (6) determining training and testing sets, (7) using random forest, gradient boosting, and naive Bayes models, and (8) evaluating the model’s performance.

### 3.1. Database Collection and Description

This research utilized two datasets collected from Taif city (a high-altitude area) and Jeddah city (a sea-level area) in Saudi Arabia for COVID-19 samples. The two datasets were employed to predict COVID-19 samples using different machine learning models, specifically the random forest model, gradient boosting model, and naive Bayes model. Between April and December 2020, a total of 1244 patients were gathered from three hospitals: King Faisal Hospital and Al-Ameen Hospital in Taif city, and the Saudi German Hospital in Jeddah city [1]. Among these, 1036 patients (83.3%) were included from Taif city (high-altitude area), and 208 patients (16.7%) were recruited from Jeddah city (sea-level area) during the same study period. The mean age of the entire patient cohort was 39.4 ± 15.9, with a male predominance of 63.3%. Most of the included patients (81.5%) tested positive for COVID-19, while 18.5% had a negative PCR result. The most common comorbidity among the patient cohort was diabetes mellitus (18.4%), followed by hypertension (14.8%), and then asthma (3.7%). The vast majority of patients complained of fever, coughing, and shortness of breath with percentages of 68%, 52%, and 49.4%, respectively. Only 32.7% of them had a known history of contact with positive cases, and 11.3% of them were healthcare workers. Most patients were admitted for less than a week, with a mean admission duration of 6 days, and 23% of them were admitted to the intensive care unit (ICU). The great majority of patients (81.2%) recovered before discharge (81.2%), while only 4.2% of patients died. Table 1 presents the baseline characteristics of the entire cohort (N = 1244).

### 3.2. Random Oversampling

The random sampling method is considered a basic strategy because it makes no assumptions about the data when applied [31]. This method entails creating a modified version of the data with a new class distribution in order to reduce the data’s inherent biases. Imbalanced data in classification become a challenge when there is a significant skew in the class distribution of the training data. This issue can significantly impact the performance of machine learning models because it tends to overlook the minority class, which is often the class of primary interest. One approach to address this problem is the random oversampling method, which involves creating new samples by randomly selecting and duplicating instances from the existing training data with replacements [32]. This process results in a newly balanced training dataset, where both classes have a more equal representation. Classification models tend to yield improved results when both classes are more evenly distributed within the dataset.

### 3.3. K Nearest Neighbor (KNN) Imputation Algorithm

KNN is a supervided learning method used for matching a point with its nearest K neighbors [33]. It can handle discrete, continuous, ordinal, and categorical data, making it very useful for dealing with missing data of any kind. The values known from the KNN algorithm are utilized to impute the missing values in the dataset instance. The closest and most comparable neighbors are identified by minimizing distance function, which is referred to as the Euclidean distance and is defined as shown in Equation (1):(1)$$E\left(a,b\right)=\sqrt{\sum_{i\epsilon D}{\left(x_{ai}-x_{bi}\right)}^{2}}$$
where $E\left(a,b\right)$ is the distance between the two instances a and b, $x_{ai}\;\mathrm{and}\;x_{bi}$ are the values of features i in instances a and b, respectively, and D is the set of attributes with non-missing values in both patterns. An important parameter for the KNN algorithm is the value of K, which refers to the number of neighbors. When the value of K is low, it increases the impact of noise in the data, and less generalizability exists in the results. As opposed to that, when the value of K is high, the distance to the donors increases, potentially leading to less precise replacement values. The recommended value for K is typically chosen as $K\approx \sqrt{N}$, where N is the number of data points. There are three ways to categorize missing data in the dataset [34]:Missing completely at random (MCAR): In this case, the dataset contains missing values that are fully independent of any observed variables in the dataset. When data is MCAR, the data analysis performed is unbiased.Missing at random (MAR): This means that missing values in the dataset are dependent on observed variables in the dataset. This type of missing data can introduce bias in the analysis, potentially unbalancing the data due to a large number of missing values in one category.Missing not at random (MNAR): In this case, missing values in the dataset are dependent on the missing values themselves and do not depend on any other observed variable. Dealing with missing values that are MNAR can be challenging because it is difficult to implement an imputation algorithm that relies on unobserved data.

### 3.4. Data Preprocessing

The data preprocessing stage holds significant importance in machine learning [35]. Data quality and preparation can greatly impact the performance of ML models during the learning process. Therefore, it is crucial to perform data preprocessing before using the data as inputs in ML models [36]. In this paper, the preprocessing stage involves normalization. When the input values in the data exhibit varying scales, normalization is applied to standardize these values [37]. Normalization is a technique used to scale the input values individually. It involves subtracting the mean (centering) and then dividing by the standard deviation. This process aims to transform the distribution of the mean value to zero and the distribution of the standard deviation value to one. The normalization process can be computed using Equation (2):(2)$$z=\frac{x-\mu }{\sigma }$$
where x denotes the input value, μ denotes the mean value, and σ denotes the standard deviation value. Mean value (μ) is computed via Equation (3):(3)$$\mu =\frac{1}{N}\overset{N}{\underset{i=1}{\sum }}x_{i}$$

Standard deviation (σ) is computed using Equation (4):(4)$$\sigma =\sqrt{\frac{1}{N}\overset{N}{\underset{i=1}{\sum }}{\left(x_{i}-\mu \right)}^{2}}$$

### 3.5. Particle Swarm Optimization (PSO)

Feature selection is the is the technique of removing irrelevant features from a large set of original features to reduce computation and reduce computation time [38]. Selecting the optimal feature subset can be challenging, and traditional methods have limitations in handling this task. To address these challenges, evolutionary computation (EC) has been proposed. One of the EC algorithms used for feature selection is Particle Swarm Optimization (PSO) [39]. In PSO, the potential solutions are referred to as particles or birds, and they do not possess volume and weight [40,41]. PSO is employed as an optimizer to find the most suitable feature subset. The best solution is found by the ith particle that is located in the D-dimensional search space. The position of particle i is presented through the vector $x_{i}=\left(x_{i1},x_{i2},\ldots ,x_{iD}\right)$, where $x_{id\in \left[l_{d},u_{d}\right]}$, $d\in \left[1,D\right]$, $l_{d}$, and $u_{d}$ present the lower bound and upper bound, respectively, of the dth dimension. The ith particle velocity is given by $v_{i}=\left(v_{i1},v_{i2},\ldots ,v_{iD}\right)$. For any particle, the best previous position is called the personal best $\left(pbest\right)$. The best solution is called the global best $\left(gbest\right)$. Random solutions are used to initialize the swarm with a population. Using pbest and gbest, the algorithm gains the best solution through updating particle velocities and positions using Equations (5) and (6):(5)$$v_{id}^{t+1}=w\times v_{id}^{t}+c_{1}\times r_{1}\times \left(p_{id}-x_{id}^{t}\right)+c_{2}\times r_{2}\times \left(p_{gd}-x_{id}^{t}\right)$$
(6)$$x_{id}^{t+1}=x_{id}^{t}+v_{id}^{t+1}$$
where t is the number of the tth iteration for the algorithm. $c_{1}$ and $c_{2}$ refer to acceleration constants. $r_{1}$ and $r_{2}$ are random values that are uniformly distributed in the range [0, 1]. $p_{id}$ is the pbest while $p_{gd}$ is the gbest. w represents the inertia weight. w provides a balance between the local search and the global search in order to improve PSO performance. v is the velocity, $v_{id}^{t+1}\in \left[v_{max},v_{min}\right]$. The boundary of the velocity is between the maximum velocity, $v_{max},$ and minimum velocity, $v_{min}$.

#### Binary Particle Swarm Optimization (BPSO)

Binary particle swarm optimization (BPSO) algorithm is a variant of particle swarm optimization (PSO) algorithm designed to handle binary optimization problems. BPSO algorithm is used for discrete problems. In BPSO, the particle position is encoded by a string that is binary. $x_{id}$, $p_{id}$, and $p_{gd}$ have values 0 or 1. The velocity of BPSO is the probability that takes value 1. The velocity is updated using Equation (5). $s(v_{id})$ is a sigmoid function to make the value of $v_{id}$ between 0 and 1. Particle position is updated by BPSO using Equation (7):(7)$$x_{id}=\left\{\begin{array}{c} 1\;\;\;\;\;\;\;\;\;\;\;\;\;\;\;\;\;\;\;\;\;\;\;if\;\;\;\;rand()<\frac{1}{1+e^{-v_{id}}}\\ 0\;\;\;\;\;\;\;\;\;\;\;\;\;\;\;\;\;\;\;\;\;\;\;\;\;\;\;\;\;\;\;\;\;\;\;\;\;\;otherwise\;\;\;\; \end{array}\right.$$
where rand() is a random number in the range 0–1. $v_{id}$ is in the range 0–1 by the sigmoid function. In BPSO, the particle is represented by binary string. When the feature mask is 1, this means that the feature is selected and otherwise, is 0. The steps of BPSO algorithm are demonstrated in Figure 2.

### 3.6. Machine Learning Models

#### 3.6.1. Random Forest Model

Random forest (RF) is a type of ensemble learning method [42] used for both classification and regression problems. RF belongs to the category of ensemble learning where multiple classifiers are combined to address complex problems and enhance the overall performance of machine learning models. In the RF model, a large number of decision trees are generated and combined. The final prediction is made through techniques such as majority voting or averaging the outputs of these individual trees. Random forest reduces variance by utilizing different samples during the training process, employing subsets of the data that contain random subsets of features, and combining the predictions of multiple smaller trees [43]. It is important to note that the decision trees in the random forest model are constructed independently. Initially, random forest may overfit the training data, but it subsequently mitigates overfitting by leveraging multiple predictors and performing averaging to achieve more robust and accurate predictions.

#### 3.6.2. Gradient Boosting Model

The gradient boosting model is a supervised machine learning algorithm used for both regression and classification problems [44]. In boosting, the fundamental concept involves combining several simple classifiers, typically derived from weak learners, to create a more powerful classifier that outperforms any single weak classifier. The gradient boosting model constructs a model based on weak learners, often using decision trees, similar to other boosting methods. However, what sets it apart is its utilization of gradients in the loss function to identify the weaknesses of these weak learners [45]. Gradient boosting employs decision trees in its process, where each decision tree’s training depends on the results obtained from the previous decision tree. This sequential and iterative nature of the training process distinguishes gradient boosting from other machine learning techniques.

#### 3.6.3. Naive Bayes Classifier Model

The naive Bayes classifier is a supervised machine learning algorithm for classification purpose [46]. In the context of the naive Bayes classifier, it assumes that all features are independent. This implies that the presence (or absence) of one particular feature in a class does not have any relation to the presence (or absence) of another feature [47]. In supervised learning, depending on the structure of the probability model, naive Bayes classifiers can be efficiently trained. The maximum likelihood strategy is used to estimate parameters for naïve Bayes models. This classifier is particularly well-suited for handling large datasets due to its simplicity and efficiency [48]. The naive Bayes classifier is well-known for outperforming even the most advanced classification algorithms due to its simplicity. It is fundamentally based on Bayes' theorem, which can be computed using Equation (8):(8)$$P\left(Z|Y\right)=\frac{P\left(Y|Z\right)\;P(Z)}{P(Y)}$$
where P(Z) is the prior probability of Z, P(Y) is the prior probability of Y, $P\left(Z|Y\right)$ is the posterior probability of Z given Y, $P\left(Y|Z\right)$ is the posterior probability of Y given Z. In Equation (8), P(Y) represents the evidence probability and no knowledge about the event Z, and Z can be true or false, then, Bayes’ theorem can be written as in Equation (9):(9)$$P\left(Z|Y\right)=\frac{P\left(Y|Z\right)P(Z)}{P\left(Y|Z\right)\times P\left(Z\right)+P\left(Y|\neg Z\right)\times P(\neg Z)}$$
where, P(¬Z) is the probability of Z that is false, $P\left(Y|\neg Z\right)$is the probability of Y given Z that is false. To define naive Bayes classifier, $X=\left\{X_{1},X_{2},\ldots ,X_{n}\right\}$ is a set of finite observed random variables, these variables are called features, and every feature has values from the domain $D_{i}$. A set of features is represented by $\varphi =D_{1}\times D_{2}\times \ldots \times D_{n}$. Assume C, where $c\in \left\{0,\ldots ,u-1\right\}$, is random variable that denotes the class of the features, and $h:\varphi \to \left\{0,\ldots ,u-1\right\}$, where h is a hypothesis that assigns the class to set of variables. Every class c assigns a function called discriminant ($f_{c}\left(x\right))$, c=0,…,u−1. The class is selected by a classifier with the discriminant function for a set of variables which is given by $h\left(x\right)=argmax_{c\in \left\{0,\ldots ,u-1\right\}}f_{c}\left(x\right)$. The discriminant function is given by Equation (10):(10)$$f\times \left(x\right)=p(C=c|X=x\;)$$

By applying Bayes’ theorem from Equation (8) to Equation (10), we obtain Equation (11):(11)$$p(C=c|X=x\;)=\frac{p(X=x|C=c\;)p\left(C=c\;\right)}{p\left(X=x\right)}$$

When $p\left(X=x\right)$ is similar across all classes, it can be ignored; so, the discriminant function can be produced by Equation (12):(12)$$f\times \left(x\right)=p(X=x|C=c\;)p\left(C=c\;\right)$$
where $p(X=x|C=c\;)p\left(C=c\;\right)$ is referred to as class conditional probability distribution. The Bayes’ classifier $h\times \left(x\right)$ is given by Equation (13):(13)$$h\times \left(x\right)=argmax_{c}p(X=x|C=c\;)p\left(C=c\;\right)$$

The Bayes’ classifier $h\times \left(x\right)$ in Equation (13) returns the maximum posterior probability for x. Equation (14), which represents the naïve Bayes’ classifier, can be used when the features are independent given the class in Equation (13):(14)$$f_{c}^{NB}\left(x\right)=\prod_{j=1}^{n}p(X_{j}=x_{j}|C=c\;)p\left(C=c\;\right)$$

### 3.7. Evaluation Metrics

Five indicators are used to evaluate the predictive capabilities of the presented prediction models and assess their performances. Each model’s performance is evaluated using metrics such as accuracy, training score, testing score, F-measure, recall, precision, and the receiver operating characteristic (ROC) curve [49]. Accuracy is computed using Equation (15):(15)$$Accuracy=\frac{TPos+TNeg}{TPos+FPos+FNeg+TNeg}$$
where TPos is true positive, TNeg is true negative, FPos is false positive, and FNeg is false negative.

Precision is calculated using Equation (16):(16)$$Precision=\frac{TPos}{TPos+FPos}$$

Recall is computed using Equation (17):(17)$$Recall=\frac{TPos}{TPos+FNeg}$$

F-measure is calculated using Equation (18):(18)$$F-\mathrm{measure}=\frac{2\times Recall\times precision}{Recall+precision}$$

Training scores and testing scores serve as indicators to assess the presence of overfitting (OF) and underfitting (UF) in machine learning models. When the training score has a high value while the testing score is low, it indicates overfitting. Conversely, when both the training score and testing score have low values, it suggests underfitting. Therefore, it is of paramount importance to avoid both overfitting and underfitting in order to achieve optimal results. The best results are typically obtained when both the training score and testing score exhibit high values. Training score and testing score are typically calculated using the "score" function in Jupyter Notebook version (6.4.6) or similar tools. These scores provide valuable insights into the model’s performance and its generalization ability to unsee data.

A binary classification model’s performance is evaluated graphically using the Receiver Operating Characteristic (ROC) curve [50]. This curve plots the true positive rate (TPR) on the y-axis and the false positive rate (FPR) on the x-axis. The ROC curve effectively illustrates the performance of a classifier across various threshold values. It allows for the identification of the threshold value where the TPR is high and the FPR is low, which is often a desirable operating point for a classifier. The ROC curve plots TPR against FPR at different thresholds. The true positive rate (TPR) represents the fraction of positive examples that are correctly classified and is calculated using Equation (19):(19)$$TPR=\frac{TPos}{TPos+FNeg}$$

The percentage of negative cases that are misclassified as positive is known as the false positive rate (FPR). Equation (20) yields FPR:(20)$$FPR=\frac{FPos}{FPos+TNeg}$$

## 4. Results and Discussion

TThe machine learning models were implemented using the Jupyter Notebook version (6.4.6). Jupyter Notebook makes creating and running Python code easier. To effectively evaluate the performance of the enhanced model in the prediction COVID-19 samples, three machine learning models were utilized for comparison. The machine learning models employed in this paper are the gradient boosting model, random forest model, and naive Bayes classifier model. In order to achieve the best performance in the prediction process, several preprocessing steps were applied:Random oversampling was used to balance the data before the training process.The KNN imputation algorithm was applied to handle missing data in the dataset.Binary Particle Swarm Optimization (BPSO) was utilized as an optimization algorithm to select important features for training, aiming for improved prediction results.

For the Jeddah city dataset, 47 features were selected out of a total 61, using the BPSO algorithm. Similarly, for the Taif city dataset, 51 features were selected out of 61 using the BPSO algorithm. The data were then normalized before being split into a training set (70%) and a testing set (30%).

In the random forest classifier model, the number of estimators was set to 100 and the same value was used for the gradient boosting model. These models were trained on the training set and evaluated using the testing set. Several evaluation metrics, including accuracy, training score, testing score, F-measure, recall, and precision, were calculated to to assess the performance of the three prediction models,

For the Taif city dataset (located in the high-altitude area of Saudi Arabia), the experimental results of accuracy, training score, testing score, F-measure, recall, and precision for the testing set are presented in Table 2 using the gradient boosting model, random forest model, and naive Bayes model, respectively.

Among all of the experimental models shown in Table 2, the gradient boosting model exhibits superior accuracy compared to the random forest and naive Bayes models. Specifically, its accuracy, training score, testing score, F-measure, recall, and precision are 94.6%, 100%, 94.5%, 94.5%, 94.6%, and 94.6%, respectively. The random forest model also performs well, with an accuracy of 93.8%, training score of 100%, testing score of 93.7%, F-measure of 93.6%, recall of 93.7%, and precision of 93.7%. In contrast, the naive Bayes model demonstrates the lowest accuracy among the three models. Its accuracy, training score, testing score, F-measure, recall, and precision are 83.3%, 87.7%, 83.4%, 83.3%, 83.3%, and 83.3%, respectively.

Figure 3 provides a visual comparison of the accuracy of the gradient boosting model, random forest model, and naive Bayes model using the dataset from Taif city in Saudi Arabia.

Figure 4 shows the ROC curve for the gradient boosting model, random forest model, and naive Bayes model, respectively, using the Taif city dataset in Saudi Arabia.

Table 3 demonstrates the results of accuracy, training score, testing score, F-measure, recall, and precision of the testing set using gradient boosting model, random forest model, and naive Bayes model, respectively, without applying random oversampling using the dataset of Taif city in Saudi Arabia.

Figure 5 demonstrates a comparison between the gradient boosting model, random forest model and naive Bayes model in terms of accuracy without applying random oversampling using the dataset of Taif city in Saudi Arabia.

From Table 2 and Table 3, the gradient boosting model gives a better accuracy than the random forest and naive Bayes models. When applying random oversampling, the accuracy of the gradient boosting model is 94.6% and without applying random oversampling, the accuracy of gradient boosting model is 85.3%; so, applying random oversampling performs better results.

For the dataset of Jeddah city (sea-level area) in Saudi Arabia, the experimental results of accuracy, training score, testing score, F-measure, recall, and precision of the testing set are shown in Table 4 using the gradient boosting model, random forest model, and naive Bayes model, respectively.

Among all the experimental models presented in Table 4, the random forest model achieves the highest accuracy compared to the gradient boosting and naive Bayes models. Specifically, its accuracy, training score, testing score, F-measure, recall, and precision are 95.5%, 100%, 95.4%, 95.5%, 95.5%, and 95.5%, respectively. The gradient boosting model also performs well, with an accuracy of 95%, a training score of 100%, a testing score of 95.1%, an F-measure of 95%, a recall of 95%, and a precision of 95%. On the other hand, the naive Bayes model demonstrates the lowest accuracy among the three models. Its accuracy, training score, testing score, F-measure, recall, and precision are 88.7%, 88.4%, 88.1%, 88.7%, 88.6%, and 88.6%, respectively.

Figure 6 provides a visual comparison of the accuracy of the gradient boosting model, random forest model, and naive Bayes model using the dataset from Jeddah city in Saudi Arabia.

Figure 7 shows the ROC curve for the gradient boosting model, random forest and naive Bayes models, respectively, using the dataset of Jeddah city in Saudi Arabia.

In terms of accuracy, the enhanced model employed in this paper yields better results for the Jeddah city dataset compared to the Taif city dataset in Saudi Arabia. Specifically, the random forest model achieves a higher accuracy with a rate of 95.5% for the Jeddah city dataset, while the best result for the Taif dataset is obtained by the gradient boosting model with an accuracy of 94.6%.

Figure 8 provides a visual comparison between the datasets of Jeddah city and Taif city in Saudi Arabia in terms of accuracy, as obtained by the enhanced model used in this paper.

Table 5 provides a comparison between the proposed work in this paper and various studies.

However, we acknowledge the importance of comparing the performance of BPSO against other algorithms to ensure the robustness of our findings. To address this suggestion, we have initiated a comparative study where we will implement and evaluate several alternative optimization algorithms, including Genetic Algorithms (GA), Simulated Annealing (SA), and Differential Evolution (DE), among others. We will analyze their performance in terms of accuracy, as shown in Table 6.

## 5. Conclusions and Future Work

In this paper, an enhanced model was developed to predict COVID-19 samples in Saudi Arabia. Two datasets were utilized, one from Taif city (high-altitude area) and the other from Jeddah city (sea-level area) in Saudi Arabia. The inclusion of these two datasets aimed to demonstrate the effectiveness and applicability of the enhanced model in different geographical areas. The enhanced model consists of eight key steps: dataset collection; using the random oversampling method for data balancing; utilizing the KNN imputation algorithm to impute missing data; data preprocessing; employing the Binary Particle Swarm Optimization (BPSO) algorithm for feature selection; selecting training and testing sets; utilizing the random forest, gradient boosting, and naive bayes classification models; and performance evaluation, including metrics such as accuracy, training score, testing score, F-measure, recall, precision, and the receiver operating characteristic (ROC) curve, calculated for each model. The COVID-19 samples were predicted using these three classification models. For the Taif city dataset (high-altitude area), the gradient boosting model proved to be the most efficient classification prediction model, outperforming random forest and naive Bayes models. Conversely, for the Jeddah city dataset (sea-level area), the random forest model yielded the best results, surpassing the gradient boosting and naive Bayes models. Notably, in terms of accuracy, the enhanced model in this paper demonstrated more effective results for the Jeddah city dataset compared to the Taif city dataset in Saudi Arabia.

Future research will explore the application of this proposed model on additional data collected from similar areas in Saudi Arabia, including high-altitude and sea-level areas. This will help identify the best prediction model for each specific region. Additionally, several machine learning and deep learning techniques will be used on these datasets to produce better forecasts for COVID-19 cases across a number of nations and locations.

## Figures and Tables

**Figure 1 biomimetics-08-00457-f001:**
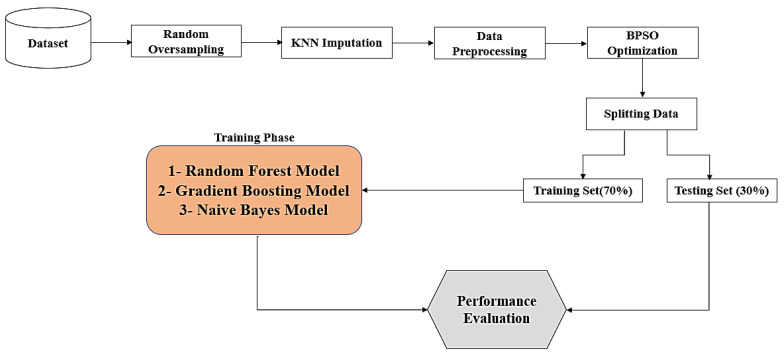
The stages for the prediction model.

**Figure 2 biomimetics-08-00457-f002:**
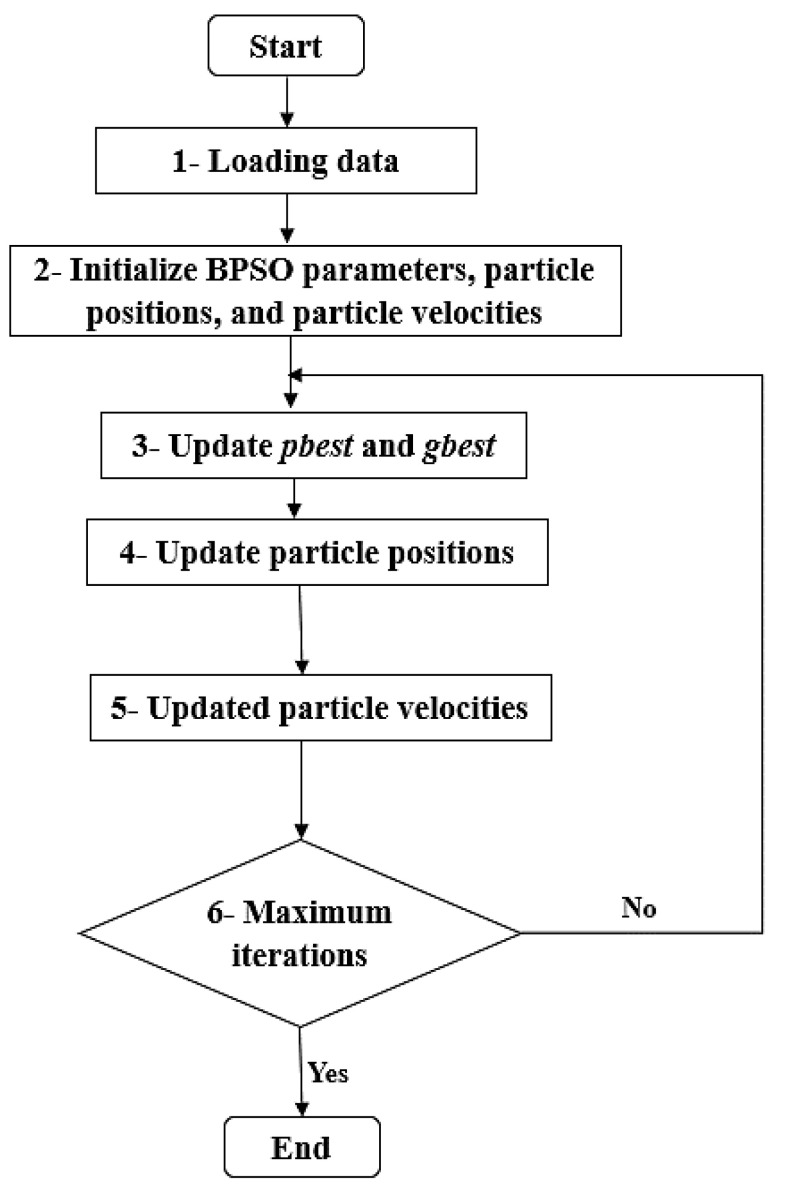
BPSO algorithm.

**Figure 3 biomimetics-08-00457-f003:**
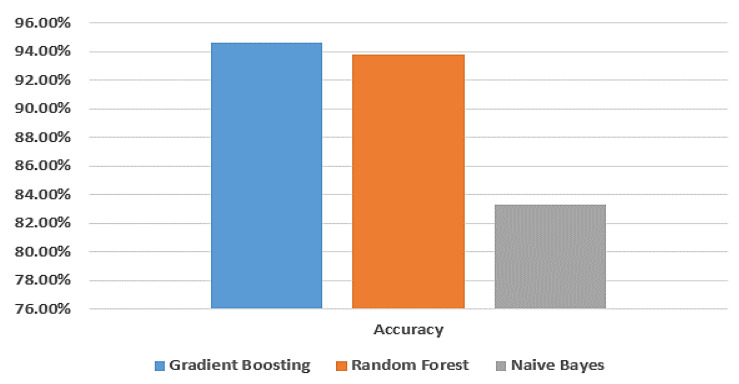
Comparison between machine learning models in terms of accuracy using Taif dataset.

**Figure 4 biomimetics-08-00457-f004:**
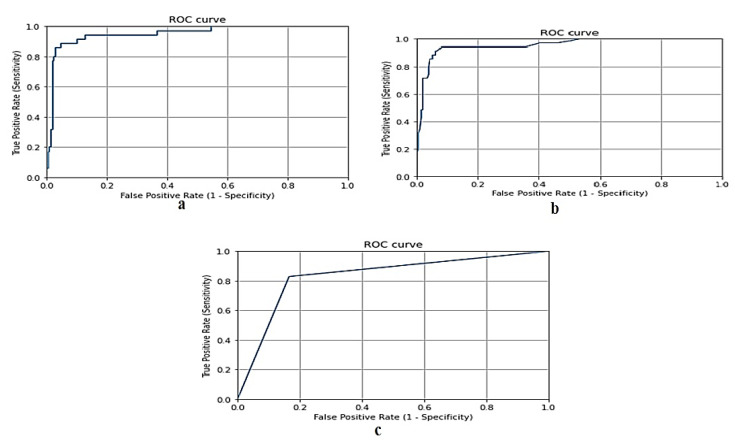
(**a**) ROC curve for gradient boosting model using Taif dataset, (**b**) ROC curve for random forest model using Taif dataset, and (**c**) ROC curve for naive Bayes model using Taif dataset.

**Figure 5 biomimetics-08-00457-f005:**
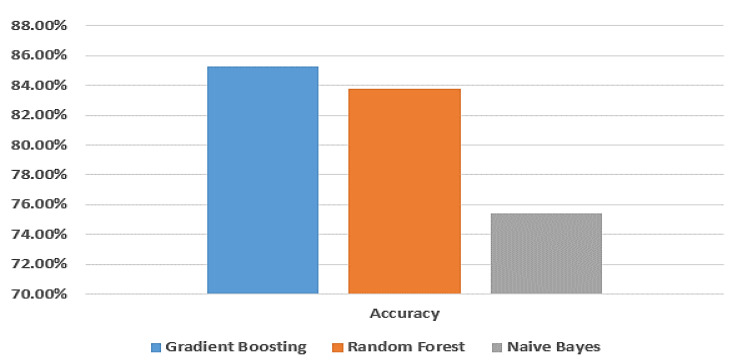
Comparison between machine learning models in terms of accuracy without applying random oversampling using Taif dataset.

**Figure 6 biomimetics-08-00457-f006:**
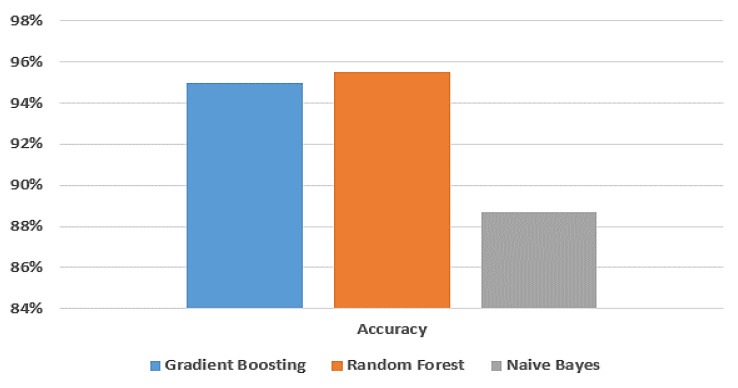
Comparison between machine learning models in terms of accuracy using Jeddah dataset.

**Figure 7 biomimetics-08-00457-f007:**
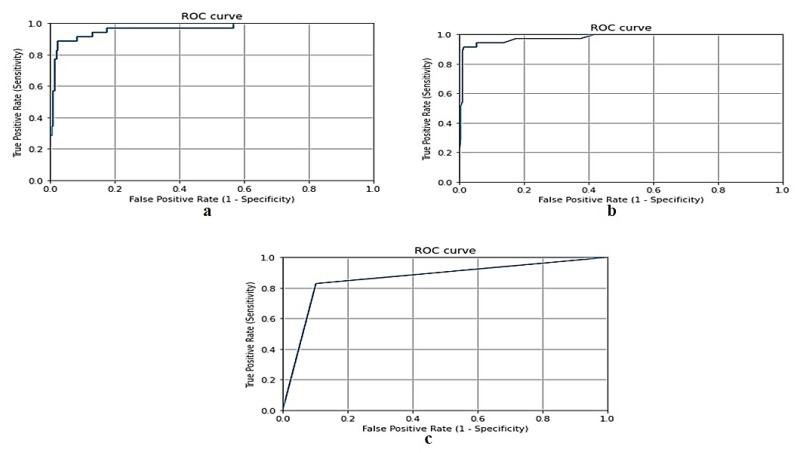
(**a**) ROC curve for gradient boosting model using Jeddah dataset, (**b**) ROC curve for random forest model using Jeddah dataset, and (**c**) ROC curve for naive Bayes model using Jeddah dataset.

**Figure 8 biomimetics-08-00457-f008:**
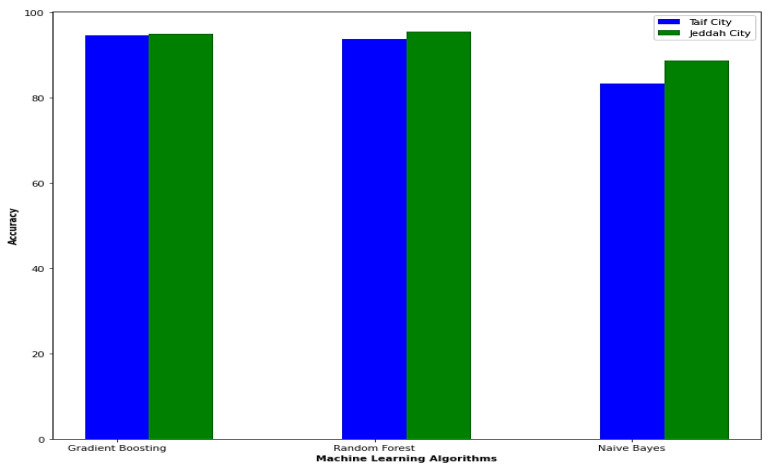
Comparison between Jeddah city dataset and Taif city dataset in Saudi Arabia in the term of accuracy.

**Table 1 biomimetics-08-00457-t001:** Baseline characteristics of the whole cohort (*n* = 1244).

Sociodemographic Data and Comorbidities			Possible Infection Source		
Mean age	39.4	±15.9	Known history of positive contact	378	32.7%
Male patients (*n*)	788	63.3%	Health care worker	127	11.3%
Female patients (*n*)	456	36.7%	Laboratory investigations		
Taif city patients (*n*)	1036	83.3%	Mean Red blood cell count (RBCs)	7.2	±2.09
Jeddah city patients (*n*)	208	16.7%	Mean Hemoglobin	15.6	±6.99
Positive COVID-19 PCR results	1013	81.5%	Mean Hematocrit	46.5	±15.63
Negative COVID-19 PCR results	230	18.5%	Mean MCV	82.8	±38.81
Diabetes mellites	228	18.4%	Mean MCH	28.5	±8.67
Hypertension	184	14.8%	Mean MCHC	31.9	±10.48
Asthma	46	3.7%	Mean RDW (%)	29.4	±66.87
Deep venous thrombosis (DVT)	7	0.6%	Mean Platelet Count	250.9	±94.23
Pulmonary embolism (PE)	10	0.8%	Mean Total WBCs	10.92	±13.46
Myocardial infarction (MI)	13	1%	Mean Neutrophil	51.56	±22.04
Ischemic stroke	20	1.6%	Mean Lymphocyte	27.70	±16.64
Acute respiratory distress syndrome (ARDS)	25	2%	Mean basophil	3.07	±9.75
Acute large vessel occlusion	3	0.2%	Mean Eosinophil	2.12	±3.00
Coronary disease	29	2.3%	Mean Monocyte	7.32	±6.05
Tumors	9	0.7%	Mean INR	4.51	±9.04
Chronic kidney disease	26	2.1%	Mean PT	16.32	±10.11
Hospital course			Mean aPTT	26.36	±13.62
Mean days of hospitalization	6.25	±6.25	Mean D-dimer	10.07	±123.48
Patient was admitted to the intensive care unit	273	23.1%	Mean ESR	28.4	±176.27
Discharged due to recovery	1009	81.2%	Mean CRP	49.8	±144.4
Discharged due to death	45	4.2%	Mean Ferritin	269.1	±359.69
Discharged upon the patients’ request (DAMA)	25	2.3%	Mean ALT	47.9	±111.75
Presenting symptoms			Mean AST	30.6	±28.31
Fever	794	68.2%	Mean ALP	66.4	±54.2
Cough	646	52%	Mean Albumin	30.9	±18.06
Shortness of Breath	575	49.4%	Mean Bilirubin	6.35	±8.03
GIT symptoms	312	26.8%	Mean Serum Creatinine test	36.18	±53.62
Headache, sore throat, or rhinorrhea	259	22.6%	Mean Blood urea nitrogen (BUN)	14.28	±22.38
Smell loss	201	20%	Mean troponin T	9.70	±12.70

**Table 2 biomimetics-08-00457-t002:** Comparison of the performances of three different machine learning models for Taif city dataset.

Models	Accuracy	Training Score	Testing Score	F-Measure	Recall	Precision
**Gradient Boosting**	**94.6%**	**100%**	**94.5%**	**94.5%**	**94.6%**	**94.6%**
**Random Forest**	93.8%	100%	93.7%	93.6%	93.7%	93.7%
**Naive Bayes**	83.3%	87.7%	83.4%	83.3%	83.3%	83.3%

**Table 3 biomimetics-08-00457-t003:** Comparison of the performances of three machine learning models without applying random oversampling using Taif dataset.

Models	Accuracy	Training Score	Testing Score	F-Measure	Recall	Precision
**Gradient Boosting**	**85.3%**	**95.2%**	**85.4%**	**85.2%**	**85.6%**	**85.6%**
**Random Forest**	83.8%	94.7%	83.7%	83.5%	83.8%	83.8%
**Naive Bayes**	75.4%	80.3%	75.3%	75.6%	75.6%	75.6%

**Table 4 biomimetics-08-00457-t004:** Comparison of prediction performances using different machine learning models using Jeddah dataset.

Models	Accuracy	Training Score	Testing Score	F-Measure	Recall	Precision
**Gradient Boosting**	95%	100%	95.1%	95%	95%	95%
**Random Forest**	**95.5%**	**100%**	**95.4%**	**95.5%**	**95.5%**	**95.5%**
**Naive Bayes**	88.7%	88.4%	88.1%	88.7%	88.6%	88.6%

**Table 5 biomimetics-08-00457-t005:** Comparison of the proposed work with a number of studies.

Studies	Model	Accuracy
**Ref [19]**	SVM	85%
**Ref [24]**	ANN	89.98%
**Ref [26]**	RF	95.03%
**Proposed Work for high-altitude area**	BPSO with gradient boosting	94.6%
**Proposed Work for sea-level area**	BPSO with random forest	95.5%

**Table 6 biomimetics-08-00457-t006:** Comparison between the performance of BPSO against other algorithms.

Model	Accuracy
BPSO	95.5%
GA	91.3%
SA	90.6%
DE	90.4%

## Data Availability

Data available on request.

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
