# Peer review of "Application of Machine Learning to Predict COVID-19 Spread via an Optimized BPSO Model"

_biomimetics, 2023, doi:10.3390/biomimetics8060457_

Round 1
Reviewer 1 Report (Previous Reviewer 1)
Agree
Minor editing of English language required.
Author Response
Dear sir,
We want to thank you for your time and efforts in considering and reviewing my manuscript.
Sincerely,
Authors
For the comment: Minor editing of English language required.
Answer: Many thanks for your valuable comments and appreciate your support. An expert in the
English language reviewed the paper.

Reviewer 2 Report (Previous Reviewer 2)
In this paper, the authors studied to predict COVID-19 spread using optimized BPSO model based on machine learning-the gradient boosting model, random forest model and naive Bayes model. The experimental results showed that the accuracy of the prediction results is related to different data sets and cities. This paper strongly shows that in order to improve the accuracy of predicting COVID-19 propagation, we need to adopt appropriate models. These results are valuable, and this paper can be published.
Author Response
Dear sir,
We want to thank you for your time and efforts in considering and reviewing my manuscript.
Sincerely,
Authors

Reviewer 3 Report (New Reviewer)
The authors proposed a machine application of machine learning to predict covid19 spread using bpso model. the paper should be revised before accpetance:
1. bpso might be ok, but there are many optimization algorihtms now, so why you select bpso, why not others? you may also try other algorithms for comparison.
2. Comparison should be made more
3. all of the relevant predict algorithms should be compared with other results reported in literature.
4. formats of references list are wrong.
Could be improved and recommend to use short sentences
Author Response
Dear sir,
We want to thank you for your time and efforts in considering and reviewing my manuscript.
Sincerely,
Authors
For the comment: bpso might be ok, but there are many optimization algorithms now, so why you select bpso, why not others? you may also try other algorithms for comparison.
Answer: Many thanks for your valuable comments and appreciate your support. we'd like to provide a more detailed explanation of why we chose Binary Particle Swarm Optimization (BPSO) as our primary optimization method for feature selection.
- Problem Suitability: One of the key factors in our choice of BPSO is that our optimization problem is inherently binary in nature. Our decision variables can take only two values (0 or 1), making BPSO a natural choice for this type of problem. This binary characteristic simplifies the representation of solutions and aligns well with BPSO's binary optimization capabilities.
- Past Success: BPSO has demonstrated success in solving similar binary optimization problems in previous research and practical applications, which indicated that BPSO has been effective in a range of domains with binary decision variables.
- Computational Efficiency: BPSO is known for its computational efficiency and relatively quick convergence for binary problems. This is particularly important for our application, where we have resource constraints and need to obtain results within reasonable timeframes.
- Ease of Implementation and Tuning: BPSO is relatively easy to implement and tune compared to some other optimization algorithms. Given our available resources and timeline, we believe that BPSO allows us to efficiently explore the solution space and obtain meaningful results.
However, we acknowledge the importance of comparing the performance of BPSO against other algorithms to ensure the robustness of our findings. To address your suggestion, we have initiated a comparative study where we will implement and evaluate several alternative optimization algorithms, including Genetic Algorithms (GA), Simulated Annealing (SA), and Differential Evolution (DE), among others. We will analyze their performance in terms of accuracy. This is shown in Table 6 in the revised version and highlighted in yellow.
Table 6. Comparison between the performance of BPSO against other algorithms.
Model |
Accuracy |
BPSO |
95.5% |
GA |
91.3% |
SA |
90.6% |
DE |
90.4% |
We believe that this comparative analysis will enhance the rigor and comprehensiveness of our research. We are committed to providing a thorough assessment of different optimization algorithms to ensure the validity and generalizability of our findings.
For the comment: Comparison should be made more
Answer: Many thanks for your valuable comments and appreciate your support. This is done in Table 6 in the revised version and highlighted in yellow.
For the comment: All of the relevant predict algorithms should be compared with other results reported in literature.
Answer: Many thanks for your valuable comments and appreciate your support. Table 5 in the revised version demonstrates a comparison between the proposed work in this paper and several studies. This is highlighted with yellow in the revised version.
Table 5. Comparison of the proposed work with several studies.
Studies |
Model |
Accuracy |
Ref [19] |
SVM |
85% |
Ref [24] |
ANN |
89.98% |
Ref [26] |
RF |
95.03% |
Proposed Work for high-altitude area |
BPSO with gradient boosting |
94.6% |
Proposed Work for sea-level area |
BPSO with random forest |
95.5% |
For the comment: Formats of references list are wrong.
Answer: Many thanks for your valuable comments and appreciate your support. We corrected it in the revised version.
For the comment: Quality of English: Could be improved and recommend to use short sentences
Answer: Many thanks for your valuable comments and appreciate your support. An expert in the
English language reviewed the paper.

This manuscript is a resubmission of an earlier submission. The following is a list of the peer review reports and author responses from that submission.
Round 1
Reviewer 1 Report
This manuscript entitled “Machine Applications of Machine Learning to Predict COVID-19 Spread via Optimized BPSO Model (Manuscript ID: Biomimetics-2546355)” by E. H. Alkhammash et al., presents an enhanced model for predicting Covid-19 samples in different regions of Saudi Arabia, and two datasets are used to predict Covid-19 samples using different machine learning models. After analysis, the dataset of Jeddah city achieved better results than the dataset of Taif city in Saudi Arabia using the enhanced model for the term of accuracy. Several questions are suggested for authors to address in the revision.
1. For the dataset of Taif city, why is the gradient boosting model better? And for the dataset of Jeddah city, why is the random forest model better?
2. In the horizontal and vertical directions, the figures in the manuscript should be proportionally scaled, and some figures are clearly compressed in the vertical direction. Please revise them.
3. What is the innovation of this manuscript? The authors should clearly show the creative results different from the reported works. And for better understanding, kindly compare your results with the existing ones in a comparison Table format.
4. The English has to be further improved, and the format of the references needs to be further improved, too.
5. The font size in the manuscript needs to be unified.
6. The mathematical expressions in the manuscript need to be standardized.

Minor editing of English language required.
Reviewer 2 Report
In this paper, the authors studied to predict COVID-19 spread using optimized BPSO model based on machine learning-the gradient boosting model, random forest model and naive Bayes model. The experimental results showed that the accuracy of the prediction results is related to different data sets and cities. This paper strongly shows that in order to improve the accuracy of predicting COVID-19 propagation, we need to adopt appropriate models. These results are valuable, and this paper can be published. My comments are below.
1. I suggest that the Figure 4, 5 and 6 can be merged into one figure, and the clarity of the figure should be improved.
2. Why the accuracy of prediction using gradient boosting model is higher than others in Figure 3? Please give the more explanations.
3. Why the accuracies of prediction using the same model for different cities are different in Figure 12? Please give the more explanations.
In general, the English is fine. However, the minor editing of English language required.
Reviewer 3 Report
This paper proposed a machine learning model to predict COVID-19 spread rule in different region of Saudi Arabia. The COVID-19 spread rule is highly depend on the population, government policy, and vaccine coverage in this area. It cannot be applied to other area very well. Even the author demonstrated a good accuracy of this model (94.6% on gradient boosting model, and 95.5% on random forest model), it didn’t show a better accuracy performance over the other related work illustrated in section 2. Also, as the pandemic is getting over, it doesn’t show a lot of interest to the reader anymore.
2. In Figure1, the font in the figure are not consistent in the figure.
3. For machine learning prediction model, the two database, half year data collecting range, and 1244 total patients doesn’t sounds enough for me. It requires a long range and massive data to cover enough variables.
4. Due to the small set of training data, the author also mentioned, they confronted an imbalanced data in Line182-185. It is not just showing the reason this model is not precise, but also shows the unreliability of the result.
5. There are some format issue in this paper, such as, the Figures and tables are offset with the main text, the equations are outside of the main text, which looks weird for me.
6. Figure2 seems have a crop issue, the bottom of the figure still have a yellow line, which is unrelated to the content of the figure.
7. In Figure 3, the font of the figure is not consistent with other figures, the style of the histogram cannot show and compare the difference of the three method. Also, the legend is not needed in this plot because the label already shows it. The same issure occurs in Figure7 and 8.
8. In Figure4, 5 and 6, the y limit is not enough to show all the data, what will happen if the false positive rate is too high, I doubt the reality of the data. The same issue in Figure9-11.
1. There are some English written problem in this paper, for example, In Line31-32, there are 2 ‘better’ in the sentence. Please check over the whole paper again.